# LSH Softmax:
# Sub-Linear Learning and Inference of the Softmax Layer in Deep Architectures

## Abstract

Log-linear models models are widely used in machine learning, and in particular are ubiquitous in deep learning architectures in the form of the softmax. While exact inference and learning of these requires linear time, it can be done approximately in sub-linear time with strong concentrations guarantees. In this work, we present LSH Softmax, a method to perform sub-linear learning and inference of the softmax layer in the deep learning setting. Our method relies on the popular Locality-Sensitive Hashing to build a well-concentrated gradient estimator, using nearest neighbors and uniform samples. We also present an inference scheme in sub-linear time for LSH Softmax using the Gumbel distribution. On language modeling, we show that Recurrent Neural Networks trained with LSH Softmax perform on-par with computing the exact softmax while requiring sub-linear computations.

## 1 Introduction

Deep neural networks have achieved impressive successes in tasks spanning vision (He et al., 2016; Krizhevsky et al., 2012), language (Bahdanau et al., 2014), speech (Graves et al., 2013; Oord et al., 2016) and videos (Abu-El-Haija et al., 2016). While these models can vastly differ in architecture, activation functions, and presence of recurrence, they (almost) all share a common trait: the softmax layer. The softmax layer, or log-linear model, is a widely used model in machine learning and statistics that transforms a feature vector into a distribution over the output space, modeling log-probabilities as a linear function of the feature vector. For example, in object classification, the softmax layer at the end of a deep convolutional network transforms a feature vector into a probability distribution over classes for the image; in language modeling using recurrent neural networks, it maps the hidden state to a distribution over next words.

While parameterizing for logits offers modeling flexibility, inference and learning have linear runtime in the number of classes. Indeed, both of these require computing the un-normalized probability for every class to compute the partition function and retrieve an actual probability distribution. Problems with large output spaces arise naturally in many areas like natural language processing (NLP), where the output space is a language's vocabulary and can be on the order of hundreds of thousands of elements Jozefowicz et al. (2016); Jean et al. (2014). This can also occur in computer vision (Joulin et al., 2016) when attempting tag prediction on massive, weakly-labeled datasets such as Flickr100M (Thomee et al., 2015).

Many solutions have been proposed to address this bottleneck, all revolving around two themes: approximation of the softmax probabilities or computation of exact probabilities for an approximate model. Canonical examples of the former are importance sampling (IS) or noise contrastive estimation (NCE; Gutmann & Hyvärinen (2012)). Instead of computing probabilities over the whole output space, these methods compute the softmax over a smaller, sampled vocabulary and re-weight the probabilities, providing an unbiased estimator. An illustration of the latter is Hierarchical Softmax (Morin & Bengio, 2005), where the output classes are first clustered such that you only need to compute the softmax over a smaller output space. While the former is an unbiased estimate, it comes with no concentration guarantees, and it is often more art than science to craft proposal distributions which will provide low-variance estimators. The latter, while efficient, requires carefully

hand-crafted clustering of the output space, at the risk of making mistakes from which there is no recovery.

More recently, estimators based on nearest neighbor search have been proposed for inference and learning in log-linear models (Mussmann & Ermon, 2016; Mussmann et al., 2017). These estimators hinge on Maximum Inner Product Search using Locality-Sensitive to retrieve the largest logits of the distribution and account for the tail with uniformly sampled classes. They boast strong theoretical guarantees and well-established concentration bounds. However, they were constrained to toy settings and not directly applicable to real-world, large-scale, machine learning. In this work, we build upon these estimators to make them amenable to deep learning practitioners, without losing any theoretical guarantees. We first show how they can be extended to be usable within training of deep learning models, then present our efficient implementation, adapted to deep learning hardware and frameworks. Finally, we show the applicability and efficiency of our method by evaluating on a real-world task: language modeling. We show significant perplexity gains against competing methods with significant speed-ups.

Our contributions are as follows:

- We present a new deep learning layer, LSH Softmax, an efficient replacement for the softmax layer based on Locality-Sensitive Hashing and the Gumbel distribution, for any deep learning architecture, with strong theoretical guarantees for sub-linear learning and inference.

- We provide details for efficient implementation on deep learning hardware (GPUs) and modern deep learning frameworks (Abadi et al., 2016; Maclaurin et al.)).

- Empirically, we show, on several datasets, that training and sampling from LSH Softmax performs similarly to an exact softmax while requiring significantly less FLOPS.

## 2 BACKGROUND

In this section, we first provide a quick overview of Neural Networks and the most popular classification layer, the softmax layer. We then present the Gumbel distribution (Gumbel & Lieblein, 1954) and introduce Locality-Sensitive Hashing (Indyk & Motwani, 1998), both of which our estimator is built upon for inference and learning. Notationally, $\mathcal{X}$ is the input space, e.g. $\mathcal{X} \triangleq \mathbf{R}^d$ and $\mathcal{Y}$ is a discrete output space: $\mathcal{Y} \triangleq \{1, \ldots, C\}$.

### 2.1 NEURAL NETWORKS

**Feedforward Networks** Neural networks models are built hierarchically by applying linear and non-linear transformations in alternating fashion. Formally, given input $x \in \mathcal{X}$, an $m$-layer neural network with $\sigma(\cdot)$ non-linearity transforms $x$ into $h$ defined as:

$$h = \sigma(W_m \cdot \sigma(\ldots \cdot \sigma(W_1 \cdot x + b_1) + \ldots) + b_m). \tag{1}$$

$\{W_i\}_{i \leq m}$ and $\{b_i\}_{i \leq m}$ are learned weights of the network. $\sigma(\cdot)$ denotes an element-wise non-linearity such as ReLU $(\max(\cdot, 0))$ or sigmoid $((1 + \exp(-\cdot))^{-1})$.

**Recurrent Networks** Recurrent Neural Networks (RNN) are an extension of the previous setting to arbitrarily long sequences by keeping an internal state $h_t$. Formally, given an input sequence $(x_1, \ldots, x_T)$, it can be written as a dynamical system of the form:

$$h_0 = \mathbf{0}; h_t = \sigma(U h_{t-1} + V x_t). \tag{2}$$

where $U$ and $V$ are learnable weight matrices. In practice, this parametrization is not well-conditioned for optimization as it can be subject to vanishing or exploding gradients and in practice the Longer Short Term Memory (LSTM; Hochreiter & Schmidhuber (1997)) is preferred.

In both cases, these outputs are then given as input to a *softmax* layer which produces a distribution over the output space $\mathcal{Y}$. In the rest of this work, we denote by $\phi$ the parameters of the neural network.

## 2.2 SOFTMAX

The softmax layer is the common name given to a log-linear model for multi-classification at the end of a neural network. Let us consider the multi-classification setting with inputs in $\mathcal{X}$ and outputs in $\mathcal{Y}$. Given a feature vector $\psi(x)$ and $C$ weight vectors $\{\theta_c\}_{c \leq C}$, the softmax layer parameterizes the following distribution:

$$p(Y = c|x; \theta) \propto \exp(\psi(x)^T \theta_c)$$

In the particular case of neural networks, $p(y|x; \theta, \phi) \propto \exp(h^T \theta_i)$. $\{h^T \theta_i\}_{i \leq C}$ are called the logits. It is important to note that computing the distribution over the output space, for inference or learning, requires $O(C)$ operations. For the rest of this work, $\theta$ denotes the parameters of the softmax whereas $\phi$ denotes the parameters of the neural network (producing the feature).

## 2.3 GUMBEL DISTRIBUTION

First introduced by Gumbel & Lieblein (1954), the Gumbel distribution is defined by the following cumulative distribution function: $p(G < s) = \exp(-\exp(-s))$. More practically, one can sample from the Gumbel distribution by first sampling $U \sim \mathcal{U}[0,1]$ and returning $G = -\log(-\log(U))$. This distribution is particularly useful as it casts sampling as optimization.

**Theorem 1** (Maddison et al. (2014)). *Let $\{y_i\}_{i \leq C}$ be un-normalized probabilities (or logits) over $\mathcal{Y}$ and let $\{G_i\}_{i \leq C}$ be i.i.d Gumbel variables. Then:*

$$\arg\max_{i \leq C}\{y_i + G_i\} \sim \text{Categorical}\left\{\frac{1}{Z}e^{y_i}\right\}_{i \leq C}$$

## 2.4 MIPS AND LOCALITY-SENSITIVE HASHING

Nearest neighbor search is a task that arises in many fields, such as information retrieval. Given a fixed set of vectors $\mathcal{S}$ and a distance, this task consists of, given any incoming query $q$, returning the vectors closest to the query according to the specified distance. In this work, we will be interested in the Maximum Inner Product Search (MIPS) task. Let $\mathcal{S} = \{s_1, \ldots, s_N\}$ be a subset of $\mathbf{R}^d$. Given a query $q \in \mathbf{R}^d$, MIPS aims at retrieving $\arg\max_{s \in \mathcal{S}} q^T s$.

This requires $\Theta(N)$ operations as one has to compute the dot-product of $q$ with all elements of $\mathcal{S}$. In the case where we assume that, for a given set $\mathcal{S}$, it is needed to retrieve the nearest neighbor for a large numbers of queries, we can achieve *amortized sub-linear time*.

This problem is commonly addressed with space partitioning techniques, such as Locality-Sensitive Hashing (LSH; Indyk & Motwani (1998)). LSH leverages hashing to reduce the number of candidate vectors to evaluate, based on the idea that similar vectors will hash in the same bucket. We have the following result:

**Theorem 2** (Indyk & Motwani (1998)). *Given a set $\mathcal{S}$ of size $N$, a similarity measure $d(\cdot, \cdot)$ and a family of hash functions $\mathcal{H}$ s.t. for $S > T$ and $p > q$:*

- $\forall x, y \in \mathcal{S}, d(x, y) \geq S \Rightarrow p[h(x) = h(y)] \geq p$.

- $\forall x, y \in \mathcal{S}, d(x, y) \leq T \Rightarrow p[h(x) = h(y)] \leq q$.

*we can construct a data structure s.t. given an incoming query $q$, a nearest neighbor can be retrieved, with high probability, in sub-linear time $O(N^\rho \log N)$ with $\rho \triangleq \frac{\log p}{\log q} < 1$.*

Recent work builds on top of LSH to either reduce the number of tables (Lv et al., 2007), or utilize more expressive hash functions (Andoni et al., 2015). A common family of hash is the *hyperplane* hash, i.e. for $v \sim \mathcal{N}(0, I), h_v(x) = \text{sign}\left(v^T x\right)$, also called Signed Random Projections (Charikar, 2002). For the rest of this work, we denote $b$ the number of hashing bits (equivalently, the number of random vectors) per table, and $L$ the number of tables.

## 3 LEARNING

In this section, we show how we can apply Theorem 3.5 of (Mussmann et al., 2017) to enable sub-linear learning of softmax parameters in the context of deep models, i.e. where both weights and inputs can change. This is crucial for real-world use.

Deep learning models for both classification (Krizhevsky et al., 2012) and generation (Mikolov, 2012) are often trained with a maximum-likelihood objective. Formally, given a training pair $(x, y) \in \mathcal{X} \times \mathcal{Y}$, one aims at maximizing $\log p(y|x; \theta, \phi)$, where $\theta \in \Theta$ and $\phi \in \Phi$ are respectively the parameters of the softmax and of the neural network. To optimize this model, the usual method is to use back-propagation (Rumelhart et al., 1988) to differentiate and then perform stochastic gradient descent (SGD; LeCun et al. (1998)) on $\theta$ and $\phi$.

Let's denote by $f(x; \phi) \triangleq h$ the feature vector given as input to the softmax. Given our notation, the objective is written as $\ell(x, y, \theta, \phi) \triangleq \log p(y|x; \theta, \phi) = h^T \theta_y - \log \sum_{i \leq C} \exp(h^T \theta_i)$. For back-propagation, we need to compute the gradient of $\ell$ w.r.t to both $\theta$ and $h$ – the gradient w.r.t. $h$ is then passed down to compute the gradient w.r.t. $\phi$.

$$\nabla_h \ell = \theta_y - \mathbb{E}_{i \sim p(\cdot|x; \theta, \phi)} [\theta_i]$$

$$\nabla_{\theta_i} \ell = \mathbf{1}_{i=y} h - h \frac{\exp(h^T \theta_i)}{Z_\theta(h)} \qquad (3)$$

with $Z_\theta(h) \triangleq \sum_i \exp(h^T \theta_i)$. Computing these gradients clearly requires $O(|\mathcal{Y}|)$ operations. In practice, this constitutes a major bottleneck for large output spaces. Mussmann et al. (2017) shows how to compute expectation in in sub-linear time, with a well-concentrated estimator using an LSH structure. Intuitively, we can build a good estimate of the partition function by retrieving the largest logits (using LSH) and accounting for the tail with uniform samples. Applying this result, we can compute the expectations necessary to compute the softmax gradients in sub-linear time. This is described in Theorem 3.

**Theorem 3** (LSH Softmax for Learning). *Let $h = f(x; \phi)$ be input to a softmax layer with parameters $\{\theta_c\}_{c \leq C}$ and define $\ell(x, y, \theta, \phi)$ as previously. Given $\mathcal{S}$, the $k$-nearest neighbors of $h$ in $\{\theta_c\}_{c \leq C}$ and $\mathcal{T}$, $l$ uniform samples from $\{1, \ldots, C\} - \mathcal{S}$, let us define:*

$$\hat{Z}_\theta(h) \triangleq \sum_{i \in \mathcal{S}} \exp(h^T \theta_i) + \frac{C-k}{l} \sum_{i \in \mathcal{T}} \exp(h^T \theta_i) \qquad (4)$$

$$\hat{g}_{\theta_i} \triangleq h \mathbf{1}_{i=y} - \left( \mathbf{1}_{i \in \mathcal{S}} + \frac{C-k}{l} \mathbf{1}_{i \in \mathcal{T}} \right) h_t \frac{\exp(h^T \theta_i)}{\hat{Z}_\theta(h)} \qquad (5)$$

$$\hat{g}_h \triangleq \theta_y - \frac{1}{\hat{Z}_\theta(h)} \left[ \sum_{i \in \mathcal{S}} \theta_i \exp(h^T \theta_i) + \frac{C-k}{l} \sum_{i \in \mathcal{T}} \theta_i \exp(h^T \theta_i) \right] \qquad (6)$$

*These estimators are well concentrated: i.e. for $\epsilon, \delta > 0$, if $k = l = O\left(n^{\frac{2}{3}} \frac{1}{\epsilon} \sqrt{\frac{1}{\delta}}\right)$, then with probability greater than $1 - \delta$:*

$$|Z_\theta(h) - \hat{Z}_\theta(h)| \leq \epsilon; \forall i \leq C, ||\nabla_{\theta_i} \ell - \hat{g}_{\theta_i}|| \leq \epsilon; ||\nabla_h \ell - \hat{g}_h|| \leq \epsilon$$

While Theorem 3 provides computation of the gradients in sub-linear time, it is only usable in a setting where the weights ($\{\theta_i\}_{i \leq C}$) are not updated. Indeed, querying nearest neighbors in sub-linear time assumes that an appropriate data structure (here LSH) was built in advance. However, when training deep models, we are required to update the weights at every training step. This necessitates online updating of the LSH structure. To maintain the sub-linear runtime, we perform these updates in a sparse manner. We describe in Algorithm 1 how this estimator can be used in a training loop, with weight updating and sparse LSH updates.

**Proposition 4.** *The softmax computations described in Algorithm 1 run in sub-linear time.*

---

**Algorithm 1** Fast Training of the Softmax layer

---

**Inputs:** Dataset $\mathcal{D} = \{(x^{(i)}, y^{(i)})\}_{i \leq N} \subset \mathcal{X} \times \mathcal{Y}$, $k$, $l$, $n_{\text{iters}}$ number of training iterations.
Initialize $\theta$ and $\phi$
Initialize the MIPS structure with $\{\theta_i\}_{i \leq |\mathcal{V}|}$.
**for** $j \leq n_{\text{iters}}$ **do**
    Sample an example $(x, y)$ from $\mathcal{D}$.
    $\Delta \theta \leftarrow 0$
    Compute $h \leftarrow f(x; \phi)$
    Find $\mathcal{S}$, $k$-nearest-neighbors of $\mathbf{h}$ using the MIPS.
    Define $\mathcal{T}$ as $l$ indexes uniformly sampled from $\mathcal{Y} - \mathcal{S}$.
    $\hat{Z}_\theta(h) \leftarrow \sum_{i \in \mathcal{S}} \exp(h^T \theta_i) + \frac{|\mathcal{Y}| - k}{l} \sum_{i \in \mathcal{T}} \exp(h^T \theta_i)$         ▷ Partition function estimate
    Output $\hat{\ell} = h^T \theta_y - \log \hat{Z}_\theta(h)$
    $\Delta \theta_i \leftarrow \Delta \theta_i + h \mathbf{1}_{i=y} - \left( \mathbf{1}_{i \in \mathcal{S}} + \frac{|\mathcal{Y}| - k}{l} \mathbf{1}_{i \in \mathcal{T}} \right) h \frac{\exp(h^T \theta_i)}{\hat{Z}_\theta(h)}$
    $\hat{g}_h \leftarrow \theta_y - \frac{1}{\hat{Z}_\theta(h)} \left[ \sum_{i \in \mathcal{S}} \theta_i \exp(h^T \theta_i) + \frac{|\mathcal{Y}| - k}{l} \sum_{i \in \mathcal{T}} \theta_i \exp(h^T \theta_i) \right]$
    Pass down $\hat{g}_{\mathbf{h}}$ for back-propagation.
    Re-hash the updated vectors (at most $(k + l)$) into the right buckets.
**end for**

---

*Proof.* The softmax computations can be split into three parts: retrieving nearest neighbors, computing the forward/backward passes, and rehashing updated vectors. With a sub-linear MIPS such as LSH, the first part is guaranteed to be sub-linear. For the second part, computing the partition function and the entire gradient estimator requires computing a finite number of sums over $O(k+l) = O(n^{\frac{2}{3}})$ terms, which is sub-linear. The third part consists of re-hashing updated vectors. Re-hashing a vector is a constant operation (consisting of $b \times L$ dot-products) and thus, given that only a sub-linear number of vectors are updated, re-hashing is sub-linear. $\qquad\square$

## 4   Inference

In the last section, we presented a method to speed-up training time based on an LSH data structure. In addition to these training time gains, LSH Softmax can be utilized for computational gains at inference time as well. While MAP inference can be easily derived from the MIPS structure, sampling from the conditional distribution is often required (e.g. to generate diverse sentences in language modeling or machine translation). These gains can be crucial for large-scale deployment. This is a direct application of (Mussmann et al., 2017) that once again leverages a MIPS structure and the Gumbel distribution. By lazily evaluating Gumbel noise, once can devise an inference scheme which allows to sample from log-linear models in sub-linear time.

**Theorem 5** (LSH Softmax for Inference). *We reuse the same notations as the once in Theorem 3. We define $t \triangleq -\log(-\log(1 - l/C))$. Let $\{G_i\}_{i \leq k}$ be $k$ samples from the Gumbel distribution. We then proceed to sample $m \sim \text{Binomial}(C, l/C)$, and sample $\mathcal{T}$, $m$ points from $\mathcal{Y} - \mathcal{S}$ with associated Gumbels $\{G_i'\}_{i \leq m}$ s.t. each $G_i'$ are larger than $t$. Let us define:*

$$\hat{y} \triangleq \arg\max \{h^T \theta_i + G_i, i \in \mathcal{S}\} \bigcup \{h^T \theta_i + G_i', i \in \mathcal{T}\}.$$

*Let $\epsilon, \delta > 0$, we then have the two following results:*

    *1. For $k = l \geq \sqrt{\log \frac{1}{\delta}}$, $\hat{y}$ is a sample from $p(y|x; \theta, \phi)$ with probability greater than $1 - \delta$.*

    *2. This inference scheme runs in sub-linear time.*

*Proof.* (Mussmann et al., 2017) $\qquad\square$

We denote by $p^{\texttt{Gumbel}}(\cdot|h; \theta)$ the implicit distribution over $\mathcal{Y}$ provided by this inference scheme. While we can sample from $p^{\texttt{Gumbel}}$, we note that the likelihood is intractable. We also emphasize that this scheme can be utilized for any softmax model, regardless of the training method.

## 5 EFFICIENT IMPLEMENTATION

Recent successes of deep neural networks hinge on their efficient implementation on specialized hardware: Graphics Processor Units (GPU), which enables training of large models in reasonable time. Often, methods with theoretically faster runtime are dismissed by practitioners because of their incompatibility with the hardware, rendering them hard to implement efficiently and ultimately not widely used. In this section, we first detail how our method is indeed amenable to GPU implementation and can amount to wall-clock gains in practice, and explain why LSH Softmax is easy to implement in the context of modern deep learning frameworks who often provide a gradient computation API.

**GPU Implementation**    Standard LSH implementations consist of three steps:

1. **Hashing**: Given a query $q \in \mathbf{R}^d$, hash $q$ into $L$ tables i.e. computing $b \times L$ dot-product with (random) hyperplanes.
2. **Look-up**: Given $L$ signatures in $\{0, 1\}^b$, retrieve candidates in each of the $L$ tables. Let us denote $C_q$ the number of candidates retrieved.
3. **Distances**: Given those candidates $\{x_1, \ldots, x_{C_q}\} \subset \mathbf{R}^d$, compute the distances $\{q^T x_i\}_{i \leq C_q}$ and only return the closest one.

It is also important to note that deep learning models are often trained using *minibatch* optimization; let us describe how each of these steps can be computed efficiently and in the minibatch setting.

The first step is amenable to the GPU setting; a batch of queries $\{q_i\}_{i \leq m} \subset \mathbf{R}^d$ can be represented by $Q \in \mathbf{R}^{m \times d}$. Given that the hyperplanes are similarly presented in matrix form i.e. $H \in \mathbf{R}^{d \times (b \times L)}$, the hashing step is equivalent to $\text{sign}\,(Q \cdot H) \in \{0, 1\}^{m \times (b \times L)}$. This is the type of operations that GPUs excel at: matrix-multiply followed by element-wise function.

The second step, while not as compatible with GPU, is still massively parallelizable using multi-threading on CPU. Given the computed signatures, one can run parallelism at the query level (i.e. each thread retrieves candidates for a given query), rendering that step efficient. It also allows for more memory-efficient look-up such as (Lv et al., 2007).

The last operation is, once again, very amenable to GPU. It simply consists of a `gather` (i.e. building a matrix with the appropriate indexes from the candidates) into a 3-d tensor. Indeed, after the previous step, the LSH structure returns $m$ lists of $s$ candidates, and the gather step returns the appropriate vectors from the vocabulary into a 3-d tensor of shape $\mathbf{R}^{m \times s \times d}$. As the batched queries can be also seen as a 3-d tensor $\mathbf{R}^{m \times d \times 1}$, computing the exact distances then reduces to a batch matrix-multiply which is a very efficient operation on GPU.

**Software Implementation**    Another crucial point for practitioners is the ability to rely on frameworks automatically providing gradients, such as (Abadi et al., 2016; Maclaurin et al.), to implement deep learning models; this abstracts away the need to write down the exact gradients which can be both cumbersome and error-prone. An additional advantage of our estimator is that it can be effortlessly implemented in these frameworks. Indeed, given logits computed over the nearest-neighbors and the additional uniformly sampled indexes, one can compute the estimate of the partition function and thus an estimate of the loss. Computing the gradient estimators now reduces to differentiating this loss, which can be very simply done using the framework's differentiation API.

## 6 EXPERIMENTS

After having presented our new layer LSH Softmax, we now proceed to show its applicability and efficiency in a real-world setting for deep learning practitioners, specifically towards language modeling. We first show that our method significantly outperforms approximate softmax baselines while performing within $20\%$ of the performance of the exact softmax. We then provide a computational comparison. While we evaluate our method on NLP tasks, we want to emphasize that it is directly applicable to other domains, such as vision. However, public vision benchmark datasets with large output spaces require significantly more computational resources (e.g. $98$ GPU nodes for $8$ days for Flickr100M (Thomee et al., 2015)) which is outside the scope of this paper.

## 6.1 Language Modeling

Language modeling is the task of, given a sequence of words $(w_1, \ldots, w_T)$ in a vocabulary $\mathcal{V}$, estimating $p(w_1, \ldots, w_T) = \prod_{t \leq T} p(w_t | w_{<t})$. Substantial work has been done to model these distributions using non-parametric $n$-gram counts with additional smoothing techniques, but can fail to model long histories because of an exponential number of sequences. Recently, parametric models using RNNs have shown impressive success on these tasks (Mikolov, 2012). In this setting, large output spaces arise naturally, as the vocabulary size can range from $10^4$ to $10^6$. We first describe our experimental protocol, and then report perplexity (ppl) of LSH Softmax against a set of baselines on this task for several datasets.

**Datasets** We evaluate our method on three standard datasets for Language Modeling with varying number of characters and vocabulary size:

- Penn TreeBank (PTB): We follow the pre-processing described by (Mikolov, 2012), which results in $929k$ training tokens, $73k$ validation and $82k$ test tokens with a $10k$ vocabulary size.

- Text8 is a dataset consisting of the first 100 millions characters of Wikipedia, and has a vocabulary size of $44k$. This dataset has been used recently in the context of language modeling (Xie et al., 2017). We use the $90M$ first words for training and split the remaining between the validation and test set.

- Wikitext-2. First introduced in Merity et al. (2016), this is a selected corpus of Wikipedia articles. It has a vocabulary size of $33k$ and contains $217k$ tokens. As previously, we split between a training, validation and testing set.

**Baselines** We evaluate the performance of models trained with (1) exact softmax i.e. computed over the entire output space, (2) Biased Importance Sampled softmax (BIS), as presented in (Jean et al., 2014), which consists of sub-sampling the vocabulary according to a proposal distribution based on unigram counts, and (3) Negative Sampling (NS), proposed in Mikolov et al. (2013), equivalent to (BIS) with a uniform distribution, (4) standard Importance Sampling (Jozefowicz et al., 2016) and (5) Noise-Contrastive Estimation (NCE; Gutmann & Hyvärinen (2012)). These baselines are what practitioners canonically use to circumvent the bottleneck of large output spaces.

**Implementation Details** Our architecture is a 2-layer RNN with LSTM cells and 650 hidden units. Weights are initialized uniformly within $[-0.1, 0.1]$. Our models are trained using SGD using gradient clipping, with an initial learning rate of 20. This learning rate is annealed when the validation perplexity plateaus. Our models are trained for 40 epochs for PTB, 3 epochs for Text8, 25 epochs for Wikitext-2. With the notations of Theorem 3, for LSH Softmax, we choose $k = 10\sqrt{|\mathcal{V}|}$ and $l = \sqrt{|\mathcal{V}|}$. For the IS and NS baselines, we choose to sample $k + l$ classes from the output space for a fair comparison. We choose the number of bits per signature $b \triangleq \log_2 |\mathcal{V}|$ and choose $L$, number of tables, to have sufficient recall for the MIPS task.

### 6.1.1 Learning

We report perplexity for a fixed architecture but comparing different softmax evaluations; we present both learning curves and perplexity on each set. We report the perplexity of all trained models using the exact probabilities i.e. the full softmax. Perplexities are reported in Table 1 and learning curves in Figure 1. We see that LSH Softmax consistently outperforms the approximate baselines by a fair margin while performing a similar number of operations, showcasing the strength of this estimator. We also observe from the training curves that approximate methods' performances tend to plateau, as IS and NS cannot target the proper classes to push down. In constrast, LSH Softmax does not.

### 6.2 Computational Comparison

Having established that the proposed estimator performs very well on real-world tasks, we now proceed to evaluate the computation gains. It is important to note that for models with large output spaces, the softmax computation can amount to about $80\%$ of the total computation (Joulin et al.,

| Method | PTB | | | Wikitext-2 | | | Text8 | | |
|--------|-------|--------|--------|--------|--------|--------|--------|--------|--------|
| | Train | Val | Test | Train | Val | Test | Train | Val | Test |
| Exact | 29.67 | 83.52 | 79.80 | 38.05 | 101.88 | 95.06 | 164.68 | 151.92 | 189.67 |
| BIS | 48.76 | 133.26 | 135.51 | 65.57 | 214.9 | 205.65 | – | – | – |
| NS | 32.12 | 103.26 | 101.48 | **42.66** | 142.82 | 136.30 | 255.62 | 234.02 | 281.11 |
| IS | – | – | 114.33 | – | – | 128.38 | – | – | **205.94** |
| NCE | – | – | 115.30 | – | – | 122.04 | – | – | 386.87 |
| Ours | **25.68** | **97.45** | **92.91** | 63.60 | **124.51** | **115.11** | **206.20** | **178.86** | 224.42 |

Table 1: LSH Softmax performs closest to the exact softmax and handily outperforms importance sampling based methods with no concentration guarantees.

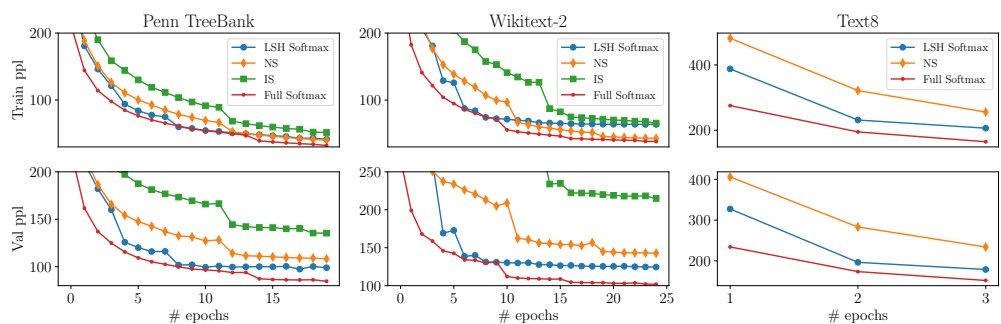

Figure 1: LSH Softmax converges faster than compared baselines on all three datasets. IS is not reported for Text8 as the results were order of magnitude worse than compared method.

2016; Ji et al., 2015); we thus choose to only evaluate computational gains in the softmax layer. We evaluate our method in CPU, with a batch size of 1, to have an accurate estimation of the ratio of FLOPS. We report both speed-up and validation perplexity (ppl) relative difference with the exact softmax for LSH Softmax and NS. Note that NS requires the same number of operations as importance sampling (IS) but outperforms it in all tasks. Additionally, we show the speed-ups one can achieve on the One Billion Word dataset (Chelba et al., 2013), whose ppl was not evaluated due to computational constraints. We report the results in Table 2. We observe that, while faster, NS performs significantly worse than LSH Softmax. Furthermore, its performance deteriorates significantly when increasing the size of the output space, contrary to LSH Softmax which always performs in the same relative range.

| Method | PTB | | Wikitext-2 | | Text8 | | Billion Word |
|--------|---------|-------------|----------|-------------|----------|-------------|--------------|
| | Speed-up | $\Delta$ ppl | Speed-up | $\Delta$ ppl | Speed-up | $\Delta$ ppl | Speed-up |
| NS | 2.8× | 23.6% | 3.7× | 40.2% | 3.1× | 54.0% | 5.7× |
| Ours | 1.6× | 16.7% | 2.4× | 22.2% | 2.3× | 17.8% | 4.1× |

Table 2: LSH Softmax performs closest to the exact softmax and handily outperforms importance sampling based methods with no concentration guarantees.

## 7 RELATED WORK

In recent years, MIPS-based estimators for log-linear models have been explored in the literature. Vijayanarasimhan et al. (2014) propose retrieving the largest logits using LSH and estimating the Softmax using only those classes. Their method is encompassed in ours by simply setting $l$ to 0. However, we note that not accounting for the tail can lead to highly biased gradients. Indeed, Mussmann et al. (2017) show that, using only the top-$k$ largest values leads to significantly worse

performance. In a similar direction, Spring & Shrivastava (2017b) propose using LSH at each layer and only retaining the largest activations which can be viewed as a form of adaptive dropout. This work differs with ours in two ways: first of all, their paper provides no theoretical guarantees and secondly, they focus on reducing memory footprint which is not the aim of our work. Finally, Spring & Shrivastava (2017a) proposed using the LSH structure as a proposal distribution to evaluate the Softmax. While unbiased and efficient, their method does not offer any concentration guarantees and the estimator can have arbitrarily bad variance.

## 8 Conclusion

In this work, we presented LSH Softmax, a softmax approximation layer for large output spaces with sub-linear learning and inference cost (in the number of states) and strong theoretical guarantees. We showcased both its applicability and efficiency by evaluating LSH on a common NLP task, language modeling. On several datasets for this task, we report perplexity closest to exact training among all baselines, as well as significant speed-ups. Our hope is that, for any architecture, this layer could be chosen in lieu of softmax, when the output space is sufficiently large to warrant the approximation. To that end, we plan to release source-code with the camera-ready version.

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
