# OpenReview forum: "LSH Softmax: Sub-Linear Learning and Inference of the Softmax Layer in Deep Architectures"
_ICLR.cc/2018/Conference — Reject_

### Official Review · AnonReviewer3 · 2017-11-26
**LSH-based methods for softmax approximation are not new, and experiments leave something to be desired**

**Rating:** 5
**Confidence:** 4

**Review:**

The paper proposes to use LSH to approximate softmax, which greatly speeds up classification with large output space. The paper is overall well-written. However, similar ideas have been proposed before, such as "Deep networks with large output spaces" by Vijayanarasimhan et. al. (ICLR 2015). And this manuscript does not provide any comparison to any of those similar methods.

A few questions about the implementation,
(1) As stated in the manuscript, the proposed method contains three steps, hashing, lookup and distance. GPU is not good at lookup, so the manuscript proposes to do lookup on CPU. Does that mean the data should go back and forth between CPU and GPU? Would this significantly increase the overhead?
(2) At page 6, the LSH structure returns m list of C candidates. Is it a typo? C is the total number of classes. And how do you guarantee that each LSH query returns the same amount of candidates?

Experiment-wise, the manuscript leaves something to be desired.
(1) More baselines be evaluated and compared. In this manuscript, only IS and NS are compared. And pure negative sampling is actually rarely used in language modeling. In addition to Vijayanarasimhan's LSH method, there are also a few other methods out there, such as hierarchical softmax, NCE, D-sothat ftmax ("Strategies for Training Large Vocabulary Neural Language Models" by Chen et. al. ACL 2016), adaptive softmax ("Efficient softmax approximation for GPUs" by Grave et. al).
(2) The results of the proposed method is not impressive. D-softmax and adaptive softmax can achieve 147 ppl on text 8 with 512 hidden units as described in other paper, while the proposed method can only achieve 224 ppl with 650 hidden units. Even the exact softmax have large difference in ppl. It looks like the authors do not tune the hyper-parameters well. With this suboptimal setting, it is hard to judge the significance of this manuscript.
(3) Why one billion word dataset is used in eval but not used for training? It is one of the best datasets to test the scalability of language models.
(4) We can see, as reported in the manuscript, that NS has bigger speedup than the proposed method. So it would be nice to show ppl vs time curve for all methods. Eventually, what we want is the best model given a fixed amount of training time. With the same amount of epochs, NS loses the advantage of being faster.

---

> ### Author Response · Authors · 2017-12-25
> **Clarifications and related work**
>
> We first and foremost want to thank you for your time and valuable comments.
>
> Comparison with Vijayanarasimhan et. al.:
> It is true that the method from this work is similar in spirit to ours. However, we wish to emphasize two key points. First of all, their method is encompassed in ours by simply setting l=0. Secondly, as shown in Mussmann et al. (UAI 2017), using the top-k largest values leads to highly biased gradients and significantly worse performance (Figure 4 and 5 of Mussman et al.).
>
> Implementation
> (1) It is important to note that the weight vectors are never copied over to CPU.
>
> The data (i.e. the weight vectors for the classes) *never* needs to be copied over to CPU. Our method only requires copying to CPU the *hashed* batch of hidden states. This consists of a bit matrix of shape (batch_size x (k * L)). This is a small matrix and thus the copying overhead is minimal.
> When copying back to GPU, one must simply copy the *indices* of the weight vectors for the gather operation, which is a small matrix (batch size x number of candidates).
>
> (2) It is a typo. We fixed that in the text, thank you for pointing it out. We guarantee a fixed number of candidates by padding with uniform samples.
>
> Experiment-wise
> (1) We added several baselines in the text, namely another (unbiased) version of Importance Sampling and NCE. We decided against comparing against Hierarchical Softmax methods (such as D-Softmax and adaptive softmax) as these requires domain-knowledge and hand-engineering. Furthermore, in contrast, they additionally enjoy no theoretical guarantees.
> (2) On Text8, our models were trained for 3 epochs, whereas the cited methods were trained for 5 or 10 epochs. Our hyperparameters were chosen from the literature for good performance with exact softmax and not tuned additionally for the approximate softmaxes.
> (3) We did not evaluate the One Billion Word dataset due to computational constraints but provided a computational comparison to show how our method could perform on even larger datasets.
>
> Thank you once again for your time and comments, we hope this addresses your concerns and that you will reconsider your rating in light of this.

---

### Official Review · AnonReviewer1 · 2017-11-27

**Rating:** 5
**Confidence:** 3

**Review:**

In this paper, the authors propose a new approximation of the softmax, based on approximate nearest neighbors search and sampling.
More precisely, they propose to approximate to partition function (which is the bottleneck to compute the softmax and its gradient), by using:
- the top-k classes (retrieved using LSH) ;
- uniform samples (to account for the tail of the distribution).
They describe how this technique can be used for learning, by performing sparse updates for the gradient (corresponding to the elements used to compute the partition function), and re-hashing the updated element of the softmax layers.
In section 5, they show how this method can be implemented on GPU, using standard operations available in neural networks framework such as TensorFlow or PyTorch.
Finally, they compare their approach to importance sampling and negative sampling, using language modeling as a benchmark.
They use 3 standards datasets to perform the evaluations: penn treebank, text8 and wikitext-2.

Pros:
 - well written and easy to read paper
 - interesting theoretical guarantees of the approximation
Cons:
 - a bit incremental
 - weak empirical evaluations
 - no support for the claim of efficient GPU implementation

== Incremental ==

While the theoretical justification of the methods are interesting, these are not a contribution of the paper (but of previous work by Mussmann et al.).
In fact, the main contribution of this paper is to show how to apply the technique of Mussmann et al. in the setup of neural network.
The main difference with Mussmann et al. is the necessity of re-hashing the updated elements of the softmax at each step.
Other previous works have also proposed to use LSH to speed up computations in neural network, but are not discussed in the paper (see list of references).

== Weak evaluations ==

I believe that the empirical evaluation of section 6 are a bit weak.
First, there is a large gap between the perplexity obtained using the proposed method and the exact softmax (e.g. 97 v.s. 83 on ptb, 115 v.s. 95 on wikitext-2).
Thus, I do not believe that the experiments support the claim that the proposed method "perform on-par with computing the exact softmax".
Moreover, these numbers are pretty far from what other papers have reported on these datasets with similar models (I am wondering if the gap would be even larger with SOTA models).
Second, the authors do not report any runtime numbers for their method and the baselines on GPUs.
I believe that it would be more fair to plot the learning curves (Fig. 1) using the runtime instead of the number of epochs.

== Efficient implementation ==

In section 5, the authors claims that their approach can be efficiently implemented on GPUs.
However, several of the operations used by their approach are inefficient, especially when using mini-batches.
The authors state that only step 2 is inefficient, but I also believe that step 3 is (compared to sampling approaches).
Indeed, for their method, each example of a mini-batch uses a different set of elements to approximate the partition function (while for other sampling methods, the same set is used for the whole batch).
Thus a matrix-matrix multiplication is replaced by n matrix-vector multiplication (n is the batch size).
While these can be performed in parallel, it is much less efficient than a matrix-matrix multiplication.
Finally, the only runtime numbers provided by the authors comparing their approach to sampling is for a CPU implementation with a batch of size 1.
This setting is super favorable to their approach, but a bit unrealistic for most practical settings.

== Missing references ==

Scalable and Sustainable Deep Learning via Randomized Hashing
Ryan Spring, Anshumali Shrivastava

A New Unbiased and Efficient Class of LSH-Based Samplers and Estimators for Partition Function Computation in Log-Linear Models
Ryan Spring, Anshumali Shrivastava

Deep networks with large output spaces
Sudheendra Vijayanarasimhan, Jonathon Shlens, Rajat Monga & Jay Yagnik

---

> ### Author Response · Authors · 2017-12-25
> **response**
>
> We first and foremost want to thank you for your time and valuable comments.
>
> Thank you for the additional references, we added those in the text along with a discussion.
>
> == ``Incremental" and related work==
>
> In addition to updating the MIPS structure with the updated weight vectors, we go above and beyond the experimental setup of Mussmann et al. 2017. Indeed, while their experiments support their theoretical results, they are far from being close to a real-world setting and usable on a large-scale task. Building on this, we extend their theoretical results and introduce LSH Softmax, which is usable in a real-world setting and on a widespread task: language modeling.
>
> Regarding the additional references: we added those in the text but we wanted to emphasize the following points:
> - Regarding (Vijayanarasimhan et al.), their method is encompassed in ours by simply setting l=0. Furthermore, as shown in Mussmann et al. (UAI 2017), using the top-k largest values leads to highly biased gradients and significantly worse performance (Figure 4 and 5 of Mussman et al.).
> - Regarding Spring et al. (KDD), there are several significant differences. First of all, their paper provides no theoretical guarantees which is a major difference with our work. Secondly, their paper focuses on reducing memory footprint which is not the aim of our work.
> - Regarding Spring et al. (arXiv), their estimator is indeed unbiased and efficient but, in contrast, provides no concentration guarantees. As with most importance sampling technique, their variance can get arbitrarily bad. Finally, the results reported on PTB are worse than those of LSH Softmax and the ones for Text8 are comparable whilst being trained for 10 epochs (theirs) compared to 3 epoch (ours).
>
> == Weak evaluations ==
>
> The gap between exact and LSH is always within 20% whilst enjoying speed-ups up to 4.1x. Regarding the exact softmax implementation on PTB, we used the hyperparameters provided by the standard PyTorch implementation. While more complex models (HyperNetworks, PointerSentinel, Variational Dropout etc...) can provide better perplexity, our baseline (79.8 on the test set) is not weak by any mean (See [1] for a thorough evaluation of various models on PTB).
>
> == Efficient Implementation ==
>
> It is true that we do not provide a GPU comparisons as our implementation is not yet competitive with TensorFlow IS and NCE implementations. However, since all of our operations are parallelizable, we posit that given professional engineering attention (which is the case for the TensorFlow IS and NCE) it should be competitive, especially given the theoretical runtime.
>
> The CPU evaluation is meant to provide us with a reasonable FLOPS estimate; on that basis, we significantly outperform competing methods.
>
> We hope that this addresses your comments and that you will reconsider your rating in light of these.
>
> [1] Regularizing and Optimizing LSTM Language Models. Merity S. et al. 2017

---

### Official Review · AnonReviewer2 · 2017-11-27
**Overall interesting idea and appealing work which is very much inline and feels like a simple addition to a previously published work**

**Rating:** 5
**Confidence:** 4

**Review:**

Authors present LSH Softmax - a fast, approximate nearest neighbor search based, approach for computing softmax that utilizes the Gumbel distribution and it relies on an LSH implementation of the maximum inner product search.

In general the work presented in this paper is very interested and the proposed method is very appealing especially on large datasets. For the most part it draws from a previous work which is my main concern. It is very much inline with the previous work by Mussmman et al. and authors don’t really do a good job in emphasizing the relationship with this work which uses two datasets for their empirical analysis. This in turn gives the overall impression that their work is a simple addition to it.

With this in mind, my other concern is that their empirical analysis are only focused on a single task from the NLP domain (language modeling).
It would be good to see how well does the model generalizes across tasks in other domains outside of NLP.
How do the different softmax approaches perform across different model configurations? It appears that the analysis were performed using a single architecture.
What about a performance comparison on an extrinsic task?
Authors should discuss the performance of LSH Softmax on the PTB train set. It appears that it outperforms the exact (i.e. “full”) Softmax or perhaps it’s an overlook on my end.

Overall it feels that the paper was written really close to the conference deadline. Given the fact that the work is mostly based on the previous work by Mussmman et al. what would make the paper stronger and definitely ready to be presented at this conference is more in-depth performance analysis that would answer some of the above questions.

LSH is typically an abbreviation for “Locality Sensitive” rather than “Locally-Sensitive” Hashing. At least this is the case with the original LSH approach.

For better clarity try rephrasing or splitting the first sentence in the second paragraph of the introduction.

I think the authors spent too much time in background section of the paper where they give an overview of concepts that should be well known to the reader (NNs and Softmax).

Theorem 3: Second sentence should be rephrased - “...and  $\mathcal{T}$, and $l$ uniform samples from…”
Theorem 3:  $\epsilon$ and $\delta$ should be formally introduced.
Section 5: pretty much covers well known concepts related to GPU implementations. Authors should spent more time focusing on the empirical analysis of their approach.

Section 6.1: “...to benchmark language modeling models...” should be rephrased.
How were the number of epochs chosen across the 3 collections?

Section 6.1.1: “...note that the final perplexities are evaluated with using the full softmax…” - This sentence is very confusing and it should be rephrased.

---

> ### Author Response · Authors · 2017-12-25
> **Clarifications**
>
> We first and foremost want to thank you for your time and valuable comments.
>
> We updated the draft to address your comments and provide more specific answers below.
>
> == Relationship with Mussmann et al. 2017 ==
>
> While Mussmann et al. 2017 provides the theoretical grounding for our work, it is important to note that their experimental setup is very constrained. While their experiments support their theoretical results, they are far from being close to a real-world setting and usable on a large-scale task. Building on this, we extend their theoretical results and introduce LSH Softmax, which is usable in a real-world setting and on a widespread task: language modeling.
>
> == Tasks from different domain than NLP ==
>
> We want to emphasize that our method is not a all domain-specific and conserves theoretical guarantees across domains. We evaluate our method on NLP task for two reasons: 1) they are particularly well-suited for evaluating our method (naturally large output spaces) 2) we did not dispose of the computational resources to tackle tasks from other domains such as vision (e.g. Flickr100M) which requires hundreds of GPUs for weeks. We briefly touched on that point in the introduction of Section 6.
>
> == Architecture and hyperparameters cross-validation ==
>
> First of all, it is important to note that our theoretical guarantees hold regardless of architecture, hyperparameters etc... Secondly, we wanted to show that our technique performed well without further parameter tuning; to that end, we tuned all of our models for the EXACT softmax. We then evaluated the approximate softmaxes by simply swapping them in without further tuning. In our opinion, ease of tuning makes these methods used in practice.
>
> == PTB Train set ==
>
> The hyperparameters (and thus regularization strength) were heavily cross-validated for performance on PTB with the EXACT softmax. It thus makes sense that the generalization gap be as small as possible in that case; it is not clear how LSH Softmax interacts with those multiple regularization schemes and thus we did not pay particular attention to that lower training perplexity.
>
> Thank you once again for your valuable comments and feedback, we hope to have addressed your concerns, and we hope you will reconsider your rating in light of this.

---

### Public Comment · ~Aaron_Jaech1 · 2017-10-29
**prior work in machine translation**

It's great that you were able to get this to work and give you a speed-up. Obviously, the amount of the speed-up you get will depend on the vocabulary size.  I would be interested in seeing more analysis of this relationship.

There was a paper from Xing Shi and Kevin Knight at ACL 2017 called "Speeding Up Neural Machine Translation Decoding by Shrinking Run-time Vocabulary" where they claim to be the first to do LSH softmax. Perhaps your implementation is better than theirs though because they didn't get a speedup. Regardless, they probably deserve some credit for trying it first.

---

> ### Author Response · Authors · 2017-10-30
> **Only addresses decoding**
>
> Thank you for the reference, we were not aware of this very recent work.  We will certainly add the citation in the next version.
>
> While related, their work only considers the case of decoding, i.e. MAP inference for a trained model and is not applicable to either learning or sampling. As we discussed in the introduction of Section 4, MAP inference is considerably easier to handle with LSH.

---

### Public Comment · (anonymous) · 2017-11-03
**Questions about baselines**

LSH Softmax seems like an interesting idea but I'm not sure what to make of the experimental results because of the weak baselines and the brittle adaptive learning rate annealing schedule.

Negative Sampling was never meant for training language models and is not an appropriate baseline here. NS is a simplification of Noise Contrastive Estimation designed for learning word representations, a task that doesn't require accurate word probabilities. Please either replace the NS results with the NCE ones or remove them altogether.

Your description of Importance Sampled softmax is incorrect, or at least incomplete, as there's more to it than just sub-sampling the vocabulary using the unigram distribution, and the reported IS results are surprisingly poor. How do you take the effect of the sampling distribution into account? What exactly is the input to the softmax? Note that both IS softmax and NCE are implemented in Tensorflow and are easy to use: https://www.tensorflow.org/versions/r0.12/api_docs/python/nn/candidate_sampling. To me, NS outperforming IS suggests that something might be seriously amiss with the experiments.

The results also seem to be highly sensitive to the learning rate annealing schedule, which is adjusted by monitoring validation error. Figure 1 suggests that one method can outperform another simply by plateauing earlier and thus inducing an earlier learning rate reduction. How was the initial learning rate chosen? For a fair comparison, you really need to do a search over the initial learning rates and annealing schedules for each method.

---

> ### Author Response · Authors · 2017-11-15
> **Additional baselines**
>
> We agree that NCE is a standard softmax approximation, however we decided (at first) not to include it because of its similarity with importance sampling (as exhibited in Jozefowicz et al. 2016) and thus it seemed redundant. We have now done the experiments and we still outperform NCE in all the cases. (see reported numbers at the end)
>
> Thank you for the link to the TensorFlow implementation. Our models were implemented in PyTorch, but our implementation follows the one presented in Jean et al. (2014) which proposes a biased partition function estimator based on a sub-sampling of the vocabulary to facilitate the matrix-matrix multiplication; this does not require reweighting of the probabilities. However, we have re-evaluated those baselines using the TensorFlow implementation, you can see the reported numbers at the end of this message. We outperform the IS TensorFlow baseline in 2 out of 3 cases and are within 10% in the third case.
>
> Regarding the remark on tuning learning rate for each method: we tuned all of our models for the EXACT softmax. Indeed, our reasoning was that we want to evaluate the approximate softmaxes by simply swapping them in without further tuning. In our opinion, ease of tuning makes these methods used in practice. This thus makes the comparison completely fair.
>
> To address the aggressive halving of the learning rate, following the standard PyTorch LM example, we halve the learning rate every time the validation loss increases (starting with lr=20); when looking at the curve, we can observe that, in no case, the learning rate get prohibitively low, which is not a reason why in some cases, plateauing could kill some methods.
>
> After your suggestions, we ran IS and NCE baselines using the TensorFlow RNNLM and their learning rate schedule (which halve at fixed epochs instead of based on validation ppl). We hereby report the results (i.e. test ppl):
> PTB:
> - IS: 114.33
> - NCE: 115.30
> - Ours (reported in paper): 92.91
> WikiText-2:
> - IS: 128.384
> - NCE: 122.041
> - Ours (reported in paper): 115.11
> Text8:
> - IS: 205.94
> - NCE: 386.87
> - Ours (reported in paper): 224.42
>
> We see that LSH Softmax still outperforms those in almost all cases, thus hopefully addressing your questions about the baselines.

---

### Public Comment · (anonymous) · 2017-11-05
**Interesting idea but results seem preliminary**

I also have some questions and comments about the baselines/experiments.

1. On text8, you report a perplexity of 190 with an exact softmax and 224 with an LSH softmax, after 3 epochs of training. The adaptive softmax paper, which you cite, reports a perplexity of 144 on text8 with an exact softmax, and 147 with an adaptive softmax, after 5 epochs of training.  The adaptive softmax also reports a larger speed up ratio on text8. I was wondering what your justification was for not comparing against the adaptive softmax (or at least mentioning this result somewhere)?

Grave et al. (2016). Efficient softmax approximation for GPUs. arXiv:1609.04309.

2. It is probably worth considering some datasets with larger vocabularies. The 44k vocab on text8 is comparatively quite small. If you do not have the computational resources to do the One Billion Word benchmark then maybe WikiText-103 or EuroParl would be reasonable choices.

---

### Public Comment · (anonymous) · 2017-11-10
**prior work in LSH + Deep Learning**

This approach draws heavily from previous LSH + Deep Learning approaches.
Yet, these papers are not cited. There should be a comparison of how this approach is novel.

 * (Vijayanarasimhan et al.) Deep Networks With Large Output Spaces (ICLR 2015)
 * (Spring et al.) Scalable and Sustainable Deep Learning via Randomized Hashing (KDD 2017)

From my understanding, this approach uses LSH to find the largest values like the previous approaches, but uses uniform random sampling to account for the tail. However, the distribution of words naturally follows the power law distribution. Shouldn't sampling from a log-uniform distribution be more computationally efficient, while providing roughly the same performance?

Grave et al. (2017) shows that "Sampled Softmax" achieves 166 perplexity on the Text8 dataset, using a single-layer LSTM with 512 units.
 * (Grave et al.) Efficient softmax approximation for GPUs. (ICML 2017)

The paper offers details on an efficient GPU implementation of their approach. However, there is not a wall-clock running time comparison against other approaches (i.e. Tensorflow Sampled Softmax) on a GPU platform. The paper compares the approaches on the basis of CPU FLOPs, which is not an accurate indicator of real-world GPU performance.

---

> ### Author Response · Authors · 2017-11-15
> **Related Work**
>
> Thanks for those suggestions, will include in the next version.
>
> Regarding (Vijayanarasimhan et al.), It is also important to note that their method is encompassed in ours by simply setting l=0. Furthermore, as shown in Mussmann et al. (UAI 2017), using the top-k largest values leads to highly biased gradients and significantly worse performance (Figure 4 and 5 of Mussman et al.).
>
> Regarding Spring et al., there are several significant differences. First of all, their paper provides no theoretical guarantees which is a major difference with our work. Secondly, their paper focuses on reducing memory footprint which is not the aim of our work.
>
> While it is true that we could hand-engineer more effective distributions for estimating the tail, our method is meant to stay as general as possible. Indeed, using uniform sampling allows our method to enjoy guarantees with no assumptions on the output distribution. Even though we only evaluated it on NLP tasks, it is effectively applicable to any domain (vision, genomics…) and thus we did not to craft an NLP-specific variant.
>
> It is true that we do not provide a GPU comparisons as our implementation is not yet competitive with TensorFlow IS and NCE implementations. However, since all of our operations are parallelizable, we posit that given professional engineering attention (which is the case for the TensorFlow IS and NCE) it should be competitive, especially given the theoretical runtime and FLOPS estimates.

---

### Decision · Program_Chairs · 2018-01-29
**ICLR 2018 Conference Acceptance Decision**

**Decision:**

Reject

**Comment:**

The authors propose an efficient LSH-based method for computing unbiased gradients for softmax layers, building on (Mussmann et al. 2017). Given the somewhat incremental nature of the method, a thorough experimental evaluation is essential to demonstrating its value. The reviewers however found the experimental section weak and expressed concerns about the choice of baselines and their surprisingly poor performance.